# CAN WATERMARKS BE USED TO DETECT LARGE LANGUAGE MODEL INTELLECTUAL PROPERTY INFRINGEMENT FOR FREE?

**Zhengyue Zhao** [1], **Xiaogeng Liu** [1], **Somesh Jha** [1], **Patrick McDaniel** [1], **Bo Li** [2], **Chaowei Xiao** [1,3]
[1] University of Wisconsin-Madison   [2] UIUC   [3] NVIDIA

## ABSTRACT

The powerful capabilities of LLMs stem from their rich training data and high-quality labeled datasets, making the training of strong LLMs a resource-intensive process, which elevates the importance of IP protection for such LLMs. Compared to gathering high-quality labeled data, directly sampling outputs from these fully trained LLMs as training data presents a more cost-effective approach. This practice—where a suspect model is fine-tuned using high-quality data derived from these LLMs, thereby gaining capabilities similar to the target model—can be seen as a form of IP infringement against the original LLM. In recent years, LLM watermarks have been proposed and used to detect whether a text is AI-generated. Intuitively, if data sampled from a watermarked LLM is used for training, the resulting model would also be influenced by this watermark. This raises the question: can we directly use such watermarks to detect IP infringement of LLMs? In this paper, we explore the potential of LLM watermarks for detecting model infringement. We find that there are two issues with direct detection: (1) The queries used to sample output from the suspect LLM have a significant impact on detectability. (2) The watermark that is easily learned by LLMs exhibits instability regarding the watermark's hash key during detection. To address these issues, we propose **LIDet**, a detection method that leverages available anchor LLMs to select suitable queries for sampling from the suspect LLM. Additionally, it adapts the detection threshold to mitigate detection failures caused by different hash keys. To demonstrate the effectiveness of this approach, we construct a challenging model set containing multiple suspect LLMs on which direct detection methods struggle to yield effective results. Our method achieves over 90% accuracy in distinguishing between infringing and clean models, demonstrating the feasibility of using LLM watermarks to detect LLM IP infringement[1].

## 1 INTRODUCTION

With the development of large language models (LLMs) and the increased scale of both model parameters and training data, substantial investments and costs have been introduced for collecting data and training the models (OpenAI, 2021; Achiam et al., 2023). Although model owners typically offer only public APIs to protect their intellectual property (IP), these models remain vulnerable: stealers can query the model's APIs to obtain high-quality data and finetune their own model with such queried data (Chiang et al., 2023; Taori et al., 2023). As a result, stealers can obtain models that perform similarly to the source LLMs in general or specialized tasks, thereby stealing the capabilities of the source LLMs and introducing IP Infringement.

To address it, a key step is to detect model IP infringement. Despite this problem having been explored in the traditional classification tasks (Tramèr et al., 2016; Jia et al., 2021; Lukas et al., 2021; Cos, 2022), IP infringement for LLMs faces two challenges. First, LLMs are designed as foundation models for diverse tasks, meaning stealers can sample data from any domain, making it difficult to determine the specific data used by the stealer. Second, the stolen data is typically used to fine-tune an existing base model rather than train a model from scratch. This introduces additional complexity in detecting model infringement, as the fine-tuned model retains the characteristics of both the original model and the new data.

---

[1] https://github.com/ZhengyueZhao/llm_infringement_detection

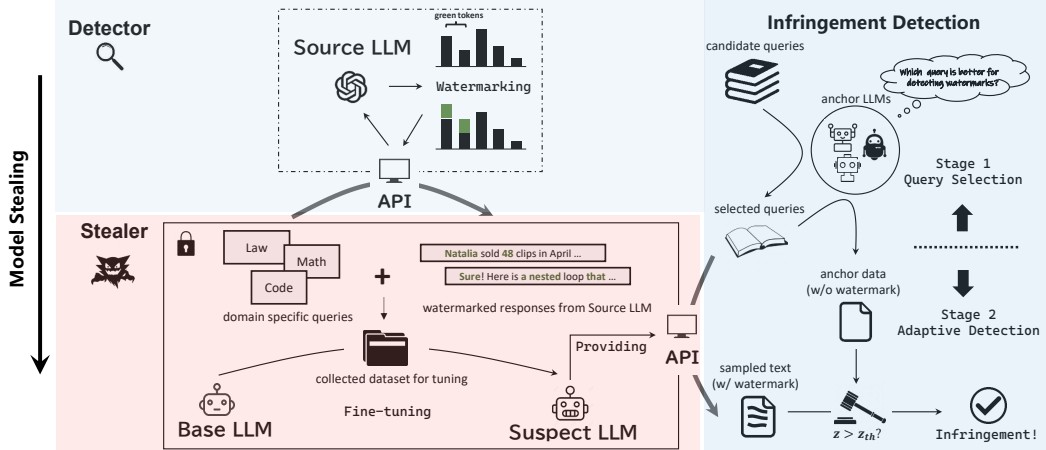

Figure 1: Overview of LLM IP infringement and its detection with decoding-based LLM watermarks. The Stealer aims to steal the source LLM by sampling with its queries and using them to tune its base LLM. While the Detector aims to detect whether a suspect LLM tuned with data sampled from the source LLM. It can access an API provided by the Stealer but does not know the base LLM and the dataset for tuning. To this end, a detection method including selecting proper queries for sampling the suspect LLM and adaptive detection with anchor data is proposed to detect IP infringement better.

An intuitive approach for model IP infringement detection is to inject a special distribution into the source model's output, such that a suspect model trained on this data would learn a special distribution, which can be used for detection, which is also known as watermarking (He et al., 2022a; Zhao et al., 2022; 2023; He et al., 2022b). Recently, watermarking has mainly been applied to detect a particular text generated by a specific LLM by injecting watermarks into the output during inference (Kirchenbauer et al., 2023; Zhao et al., 2024; Kirchenbauer et al., 2024a; Kuditipudi et al., 2024). This watermarking technique works by dividing the vocabulary into a "green list" and a "red list", and adding bias to the logits of different tokens. This increases the frequency of tokens from the green list in the output text so that the watermarked text can be detected with a fixed threshold of z-score from a z-test of the green ratio. Since watermarks can change the distribution of output, a natural question emerges: "Could such watermarks be used to detect LLM model infringement for free?"

In this paper, we propose a watermark-based LLM model infringement detection algorithm, **LIDet** (**L**LM **I**nfringement **Det**ection). Our journey begins by identifying the key challenges of using LLM watermarks for model infringement detection, compared to detecting generated text.

The first challenge is ***attenuation of watermark detectability*** *caused by* ***domain mismatch*** *between training and detection*. As LLMs are foundation models designed for diverse domains, stealers can extract data from arbitrary domains to fine-tune their suspect models, such as code generation and mathematics. The stealer can avoid being identified by accessing the API of the target model using anonymous accounts. Consequently, the detector is unaware of the specific domain data used by the stealer. Thus, the detector needs to construct detection data. However, if the detection data has a large distribution mismatch with the training data, it leads to an attenuation in the frequency of watermarked tokens in the text, weakening detection capability, as shown in Figure 2a.

The second challenge is *the guarantee of watermark learnability results in a* ***green ratio mismatch***, *leading to the failure of watermark detection for some hash keys*. Unlike watermarking for detecting generated text, where the watermark is directly injected, watermarking for suspect LLM infringement detection involves the watermark being learned through the training process on unknown sampled data. The detectability of the watermark in the suspect model is thus highly dependent on its learnability. To ensure the watermark embedded in the source model is also learned by the suspect model, careful design of the watermarking algorithm is essential. While text watermarking typically uses high n-gram lengths (e.g., 4) to ensure robustness against reverse engineering (Kirchenbauer et al., 2024b), this reduces the learnability of the watermark (Gu et al., 2024a), as also shown by our experiments in Figure 2b. To address this, an intuitive approach to improve the learnability

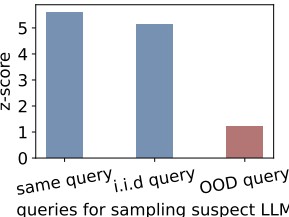 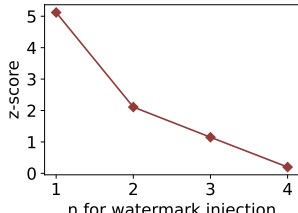 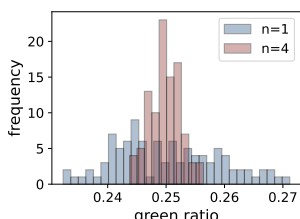

(a) Queries for detection. Same & i.i.d queries are in-domain while OOD queries are out-of-domain queries of queries for tuning.

(b) Hyper-parameter $n$ of KGW watermark: the hash for green list is calculated with a hash key and previous $n$ tokens' ids.

(c) Histogram of actual green ratios of KGW watermark with 100 random hash keys.

Figure 2: (a) and (b) show z-scores of text sampled from the suspect LLM tuned with KGW-watermarked data. A higher z-score represents more watermarked green tokens in the suspect LLM, indicating better learnability of watermarks and detectability of suspect LLMs. (c) demonstrates greater $n$ leads to the actual green ratio being unstable from the preset green ratio 0.25.

of the text watermark is to decrease the n-gram length, for example, setting it to 1. However, this introduces another problem: using a low n-gram length causes the actual green ratio to be heavily influenced by the random hash key, as shown in Figure 2c. This mismatch between the actual green ratio and the preset green ratio for partitioning the green list makes the z-score-based detection with a fixed threshold fail, which means that even with a valid text watermarking scheme like KGW (Kirchenbauer et al., 2023), the watermark may be detectable with some hash keys but not with others.

To address these challenges, our proposed method, **LIDet**, contains two designs as briefly demonstrated in Figure 1: (1) To mitigate the attenuation of detectability introduced by mismatched training and detection domains, we apply several anchor LLMs (which can be different from the suspect LLM) to help select such queries that are more likely to sample texts containing more diverse and evenly distributed tokens. Compared with those queries that bring texts with highly repetitive or centrally distributed tokens, these queries could get responses containing effectively watermarked tokens with higher probability. (2) To solve the mismatched green ratio brought by watermarks with a low $n$ for better learnability, we replace the fixed threshold of z-score-based detection with an adaptive threshold by estimating an actual green ratio from pre-generated anchor data. Meanwhile, we take the cross-model generalization into account during query selection to ensure the green ratio of anchor data generated by anchor LLMs can be closer to the natural green ratio of un-watermarked text. Therefore, the reference green ratio can help better discriminate between the infringed suspect LM and the clean one.

We conduct experiments on a black-box scenario from the detector's perspective with LLM watermarks such as KGW (Kirchenbauer et al., 2023) and Unigram (Zhao et al., 2024), as well as different source LLMs, suspect LLMs, and datasets for tuning suspect LLMs. Results reveal that our proposed detection method increases the reliability of discriminating against unauthorized distilling of source LLMs and further demonstrates successful cases of using LLM watermarks for defending against LLM's model infringement. Specifically, our method achieves the detection accuracy of over 90% in the cross-domain detection on a challenging model set containing suspect LLMs with multiple settings while the vanilla detection struggles to provide effective results.

Our contributions can be summarized as follows:

- We explore the feasibility of using existing LLM watermarks to detect model infringement and analyze the limitations of vanilla detection methods in this task.

- We propose a simple yet effective detection method, **LIDet**, to enhance LLM model infringement detection, significantly improving the detectability of model infringement.

- Through extensive experiments under realistic threat models, we demonstrate that LLM watermarks with our proposed detection methods, can effectively detect model infringement and thus protect the copyright of source LLMs.

## 2 BACKGROUND & PRELIMINARY

### 2.1 LLM WATERMARKS AND DETECTION

**KGW.** KGW (Kirchenbauer et al., 2023) modifies the frequency of certain tokens in the generated text by adding a bias to the logits of specific tokens during LLM decoding. Specifically, during each decoding step, KGW randomly divides the tokenizer's vocabulary table into a green list and a red list based on a predefined green list ratio $\gamma$ and hash key $\xi$ using a hash function applied to the preceding $n$ tokens: $f_{\text{hash}}(\xi, x_{-1}, ..., x_{-n})$. Logits of tokens in the green list are increased by a fixed hyperparameter $\delta$, thus raising the probability that these green tokens will be sampled during decoding. Detection of KGW works by statistically analyzing the proportion of green list tokens in the target text (with $|T|$ tokens in total). After determining the number of green tokens $|s_G|$ in the target text, a z-test is performed to compute the z-score or p-value to evaluate whether the proportion of green tokens $|s_G|/|T|$ significantly exceeds the preset green ratio $\gamma_0$: $z = (|s_G| - \gamma_0|T|)/\sqrt{\gamma_0(1-\gamma_0)|T|}$. Typically, if the z-score exceeds a predetermined threshold $\alpha$ (such as 4.0), the text is considered to be watermarked.

**Unigram.** The generation and detection of the Unigram (Zhao et al., 2024) watermark are similar to KGW. The key difference is that Unigram does not use the preceding n tokens to compute the hash function; instead, it divides the green list solely based on the hash key $\xi$. As a result, the green list of Unigram is static. Compared to KGW ($n \geq 1$), Unigram ($n = 0$) significantly enhances the robustness and detectability of the text watermark. However, this also increases the influence of the hash key on green tokens, making the actual green ratio deviate from the preset green ratio $\gamma$.

### 2.2 PROBLEM STATEMENT

The process of LLM model stealing and infringement detection can be divided into the following stages as illustrated in Figure 1: (1) The stealer constructs a query set $\mathbb{Q}^T$ intended for fine-tuning their base model. By accessing the source LLM's API, they sample the corresponding responses $\mathbb{R}^T$ from the query set, thus creating a dataset $\mathbb{D}^T$ for instruction fine-tuning. (2) The stealer applies the constructed training dataset $\mathbb{D}^T$ to fine-tune their own base model, resulting in a suspect LLM $\theta^{\text{suspect}}$. They then provide the API of the suspect LLM. (3) The detector accesses the API of the suspect LLM $\theta^{\text{suspect}}$ and uses a series of queries $\mathbb{Q}^D$ to sample output text $\mathbb{R}^D$ from the suspect LLM's API. (4) The detector analyzes the concatenated output text $\mathbb{R}^D$ from the suspect LLM $\theta^{\text{suspect}}$ to check for the presence of the source LLM's watermark, thereby determining whether the suspect LLM is fine-tuned using data sampled from the source LLM.

### 2.3 THREAT MODEL

We consider two opposing roles in the task: the Detector (the owner of the source LLM and the entity responsible for detecting infringement) and the Stealer (the owner of the suspect LLM).

**Stealer.** The Stealer is the owner of the suspect LLM, who attempts to steal the capabilities of the source LLM by fine-tuning their base model using data sampled from the source LLM. The Stealer can only sample data through the source LLM's API but has the flexibility to choose any query for sampling. We consider the realistic scenarios where the Stealer can use different IPs and anonymous accounts to query the source LLM so that the detector can not know what data are queried. In the context of stealing a target LLM, we treat this fine-tuning as the last training operation of the suspect LLM. The Stealer would also not try to paraphrase the sampled text considering the paraphrasing will change the stealing target from the source LLM to the paraphrasing model. After fine-tuning, the Stealer provides the API of the suspect LLM to the public, without revealing the model structure or training data.

**Detector.** The Detector is the owner of the source LLM and aims to determine whether the suspect LLM has infringed upon the source model. The Detector has full knowledge of the source LLM, including model weights and watermarking configurations. However, during detection, the Detector can only access the suspect LLM's API and is unaware of its model structure or the data used for training. The Detector must rely solely on the output text generated by the suspect LLM to determine if it has infringed upon the source LLM.

Table 1: Detectability of suspect models with different queries. A higher z-score or a lower p-value indicates that the suspect LLM is more likely to be watermarked.

| Detection Query | Watermark | w/ watermark | | w/o watermark | |
|---|---|---|---|---|---|
| | | z-score | p-value | z-score | p-value |
| Training Query | Unigram | 27.6 | 1.4e-167 | 0.731 | 2.3e-1 |
| | KGW | 5.61 | 1.04e-8 | 0.406 | 3.4e-1 |
| i.i.d Query | Unigram | 24.6 | 4.9e-134 | 0.534 | 3.0e-1 |
| | KGW | 5.12 | 1.5e-7 | -0.281 | 6.1e-1 |
| OOD Query | Unigram | 11.2 | 3.5e-29 | -3.67 | 9.9e-1 |
| | KGW | 1.21 | 1.1e-1 | -1.79 | 9.6e-1 |

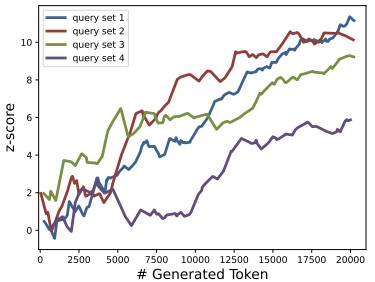

(a) Unigram-watermarked suspect LLM.

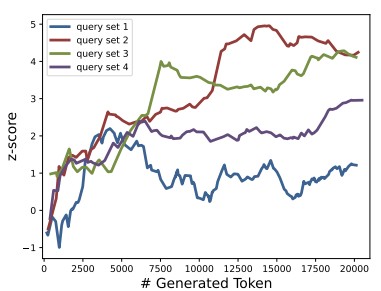

(b) KGW-watermarked suspect LLM.

Figure 3: Z-scores of generated text sampled with different out-of-domain query sets.

# 3 METHODOLOGY

The difference between the directly injected text watermark and the learning-based LLM's model watermark brings two challenges that hurt the effectiveness of detecting infringing suspect LLMs: (1) The scenario where the detector can not get training data of the suspect LLM means there is a domain mismatch between training and detection data, resulting in the **attenuation of watermark detectability**. (2) The demand for learnable watermarks limits the $n$ of watermarks to small, leading to **green ratio mismatch** between the actual one and the preset one, which further causes failed detection. We first discuss these two challenges and then introduce our **LIDet** for reliable detection to address them.

## 3.1 ATTENUATION OF WATERMARKS DETECTABILITY IN DOMAIN MISMATCH

The detection of a suspect LLM involves two key steps: sampling text from the suspect LLM for detection, and detecting the watermark in the text. In this case, an important consideration is to check if the watermark is detectable under the threat model, i.e. whether watermarks of suspect LLMs can be detected without knowledge of the training data. To answer this question, we conduct a preliminary experiment to first fine-tune a suspect LLM with **coding** data (Luo et al., 2024) sampled from the source LLM. Then three different query sets are applied to sample text from the tuned suspect LLM: the same queries for training the suspect model (the first-row of Table 1), queries different from training queries but in the same domain (also coding queries, the second-row of Table 1), and the out of domain queries (general queries sampled from Alpaca (Taori et al., 2023), the third-row of Table 1).

Results from Table 1 reveal that the detectability of LLM watermarks attenuates along with the increase of difference between detection queries and training queries. The z-score of the sampled text is significantly lower compared to when the detector knows the suspect LLM's training queries, indicating a reduced detection capability. Intuitively, this occurs because the watermark in the suspect LLM is learned from the training data, so the closer the sampled text is to the training data during detection, the better the detection performance.

Figure 3 further illustrates the detectability of different out-of-domain detection queries. It indicates that though the detected texts are all sampled with general query sets, the detectability of these cases varies. The curve of text sampled with less-detectable queries (e.g. query set 1 of Figure 3b) shows that the z-score sometimes significantly drops while the number of tokens increases, which

Table 2: Learnability of watermarks with different $n$. A higher z-score or a lower p-value indicates that the watermark is more learnable during the tuning process of the suspect LLM.

| Watermark | | w/ watermark | | w/o watermark | |
| --- | --- | --- | --- | --- | --- |
| | | z-score | p-value | z-score | p-value |
| Unigram | n=0 | 24.6 | 4.9e-134 | 0.534 | 3.0e-1 |
| KGW | n=1 | 5.12 | 1.5e-7 | -0.281 | 6.1e-1 |
| | n=2 | 2.11 | 1.7e-2 | -0.196 | 5.8e-1 |
| | n=3 | 1.15 | 1.3e-1 | 0.056 | 4.8e-1 |
| | n=4 | 0.20 | 4.2e-1 | -0.061 | 5.2e-1 |

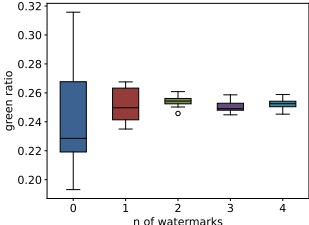 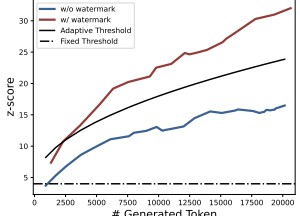 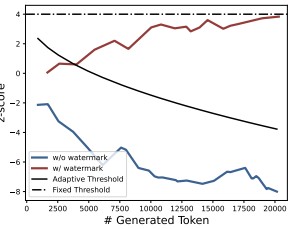

(a) Distribution of actual green ratio (b) z-scores with a higher green ratio. (c) z-scores with a lower green ratio. with $\gamma = 0.25$ with varies hash keys.

Figure 4: A small $n$ can lead to failed detection. (a) a small $n$ makes the actual green ratio deviate from the set of 0.25. (b) z-score of unwatermarked text easily surpasses the fixed threshold when the actual green ratio is higher than $\gamma$. (c) z-score of watermarked text is smaller than the fixed threshold when the actual green ratio is lower than $\gamma$.

demonstrates that some queries make negative contributions to detecting watermarks. In contrast, query sets such as 2 and 3 from Figure 3b still demonstrate enough z-scores for watermark detection. We find that those queries causing z-score dropping usually have low diversity responses in tokens, which are short in length or have high repetition tokens, thus leading to a bad detection. Therefore, it is crucial to screen out unhelpful queries from an out-of-domain general query set to detect LLM infringement without in-domain queries.

## 3.2 GREEN RATIO MISMATCH OF LEARNABLE WATERMARKS

The learnability of LLM watermarks decides whether watermarks of source LLMs' outputs can be transferred to the suspect LLM during fine-tuning. Obviously, the learnability of watermarks is highly related to the hyper-parameter $n$ in KGW. As shown in Table 2, a watermark with a greater $n$ results in a smaller z-score and higher p-value, which means it is less learnable for suspect LLMs. Intuitively, when the watermark is transferred from the source LLM to the suspect LLM, the suspect LLM essentially learns the distribution shift introduced by the n-gram-based watermark. When $n$ increases, the distribution of n-grams in the training data becomes more dispersed, making it more difficult for the suspect LLM to sufficiently learn the n-gram watermark with the same amount of training data. This implies that smaller values of $n$ ($n = 0, 1$) are more effective for detecting model infringement, as they make it easier for the watermark to be learned and subsequently detected.

However, the learnable watermark with small $n$ leads to another problem: the partition of the green list is rather sensitive to the random hash key $\xi$. Figure 4a illustrates the distribution of the actual green ratio of unwatermarked texts with random hash keys, which reveals that watermarks with small $n$ can result in a mismatch between the actual green ratio and the preset green ratio $\gamma$ during partition. Considering that the calculation of the z-score for detection is dependent on the green ratio $\gamma$, such a mismatch could be disastrous for watermark detection. When the actual green ratio is lower than the set $\gamma$, the calculated z-score will be lower than the actual value. In this case, the z-score of text with the watermark might fall below the detection threshold, leading to a higher false negative rate (as shown in Figure 4c). Conversely, when the actual green ratio is higher than the set $\gamma$, the calculated z-score will be higher than the actual value. This can result in even non-watermarked text having a z-score above the detection threshold (as shown in Figure 4b), thereby increasing the false positive rate.

These results highlight a trade-off between the learnability of LLM watermarks and the stability of the green ratio with respect to the hyper-parameter $n$ when detecting model infringement. To

ensure the watermark's learnability and detectability together, it is crucial to replace the original fixed threshold of z-scores with an adaptive threshold which can compensate for the instability of the actual green ratio brought by the small $n$.

### 3.3 LIDET: RELIABLE DETECTION FOR LLM INFRINGEMENT

Our proposed **LIDet** includes two key stages: (1) selecting queries for sampling detection text from the suspect LLM with the help of a set of anchor LLMs, and (2) detecting watermarks from the detection text with an adaptive threshold of z-score. We address the challenge introduced by domain mismatch by selecting queries that can promote responses with diverse and frequency-balanced tokens, regarded as the principle of **token entropy**. Besides, we mitigate the problem caused by mismatched green ratios via selecting cross-model generalizable queries and detecting the watermark with an adaptive threshold to fill in the z-score gap by estimating the actual green ratio.

**Anchoring Treated Query Selection.** The queries are selected by two principles: (1) the **token entropy** of their responses, and (2) their **cross-model generalizability**. To identify the property of each candidate query, we introduce a set of anchor LLMs $\{\boldsymbol{\theta}_{\text{anchor}}\}^M$ first. These anchor LLMs are used to help select a proper query set $\mathbb{Q}^D$ from all candidates $\mathbb{Q} = \{\boldsymbol{q}_1, \boldsymbol{q}_2, ..., \boldsymbol{q}_K\}$. By sampling responded text with all $K$ candidate queries from $M$ anchor LLMs, a set of anchor responses $\mathbb{R}^D$ is constructed:

$$\mathbb{R}^D = \bigcup_{m=1}^{M} \bigcup_{k=1}^{K} \boldsymbol{\theta}_{\text{anchor}}^m(\boldsymbol{q}_k \in \mathbb{Q}) = \{\boldsymbol{r}_1^1, ..., \boldsymbol{r}_k^m, ..., \boldsymbol{r}_K^M\} \tag{1}$$

Based on the anchor responses, we define the **token entropy** $H$ of the response for each query $\boldsymbol{q}_k$ as the frequency balance degree of contained tokens across all anchor LLMs, shown in equation 2, where $\tau$ refers to tokens in the corresponding response $\boldsymbol{r}_k^m$ and $P(\tau)$ stands for the frequency of token $\tau$. Query with a higher $H$ usually can promote a response containing more tokens and the distribution of tokens is more even, making it more likely to include watermarked tokens.

$$H(\boldsymbol{q}_k) = -\mathbb{E}_{m=1,...,M} \left[ \sum_{\tau \in \boldsymbol{r}_k^m} P(\tau) \log P(\tau) \right] \tag{2}$$

Then we introduce the concept of the **cross-model generalizability** $\eta$ of queries. Specifically, $\eta$ of query $\boldsymbol{q}_k$ is defined as the negative variance of the statistical green ratio $\gamma_k^m$ of the corresponding response $\boldsymbol{r}_k^m$ sampled from all $m$ anchor LLMs, as demonstrated in equation 3. The green ratio $\gamma_k^m$ of response $\gamma_k^m$ is calculated by the number of green tokens $|\boldsymbol{s}_G|_k^m$ and the number of all tokens $|\boldsymbol{r}_k^m|$. A higher $\eta$ indicates the green ratio of the response sampled from different anchor LLMs is closer, which means that the corresponding query is more likely to produce texts with similar green ratios across various LLMs. Therefore, we could avoid queries that yield significantly different green ratios when sampled from different models by selecting queries with smaller $\eta$, thus providing a better estimation of the actual green ratio of the suspect LLM in the next stage.

$$\eta(\boldsymbol{q}_k) = -\text{Var}_{m=1,...,M}(\gamma_k^m), \quad \gamma_k^m = |\boldsymbol{s}_G|_k^m / |\boldsymbol{r}_k^m| \tag{3}$$

According to the token entropy $H$ and cross-model generalizability $\eta$ of all candidate queries, we select detection queries following the priority score $v$ given in equation 4. Queries with higher token entropy and higher cross-model generalizability are selected first to sample the suspect LLM to get detection texts.

$$v(\boldsymbol{q}_k) = \frac{H(\boldsymbol{q}_k) - \min H(\boldsymbol{q}_k)}{\max H(\boldsymbol{q}_k) - \min H(\boldsymbol{q}_k)} + \frac{\eta(\boldsymbol{q}_k) - \min \eta(\boldsymbol{q}_k)}{\max \eta(\boldsymbol{q}_k) - \min \eta(\boldsymbol{q}_k)} \tag{4}$$

**Detection with Adaptive Threshold.** By the above stage of query selection, the query set $\mathbb{Q}^D$ is built. The texts for watermark detection are sampled from the suspect LLM with $\mathbb{Q}^D$. Specifically, we sample corresponding responses from the suspect LLM with each query in $\mathbb{Q}^D$ and then concatenate them together for the later detection process. For z-score-based detection, instead of applying the fixed threshold $\alpha$, we propose a watermark-adapted threshold $z_{th}$ to correct the mismatch between the actual green ratio and the preset green rate $\gamma_0$. The adaptive threshold works by filling in the gap of z-scores caused by the mismatched actual green ratio. To estimate the actual green ratio, we collect responses sampled from anchor LLMs with $\mathbb{Q}^D$ as **anchor data**. The statistical green ratio $\mu_G$ of the anchor data is treated as an approximation of the actual green ratio.

$$z_{th} = \frac{\mu_G - \gamma_0}{\sqrt{\gamma_0(1 - \gamma_0)}}\sqrt{|T|} + \alpha \qquad (5)$$

Then the adaptive threshold can be obtained by equation 5, where $|T|$ is the number of tokens in the text for detection. Figure 4b and Figure 4c demonstrate the curve of adaptive threshold $z_{th}$ as well as the fixed threshold $\alpha$. It is clear that while the actual green ratio mismatches with the green ratio for partition, $\alpha$ fails to discriminate the watermarked and clean model but $z_{th}$ is successful in classifying the two models.

In summary, the reliable detection of the suspect model with **LIDet** contains the following processes: (1) sample the anchor models with all queries to get corresponding responses; (2) calculate green ratio $\gamma$ and token entropy $H$ of the response of each query $q_k$ and anchor model $\boldsymbol{\theta}_{\text{anchor}}^m$; (3) calculate the priority score $v$ of each query $q_k$ following equation 4; (4) sample text from the suspect LLM with queries of higher priority $v$ and then calculate $\mu_G$ of equation 5 with anchor data (text sampled from anchor models with the same queries); (5) concatenate the sampled text and then calculate z-score of all sampled text by $z = (|s_G| - \gamma_0|T|)/\sqrt{\gamma_0(1 - \gamma_0)|T|}$ and calculate $z_{th}$ following equation 5. If $z > z_{th}$, then the suspect LLM is considered an infringing model.

Though **LIDet** requires the detector to sample all anchor LLMs using a query set to obtain the outputs corresponding to all queries, the resulting query subset and green ratio are applicable to any suspect LLM. As long as the watermark configuration of the source LLM is determined, there is no need to re-filter the query subset when detecting new suspect LLMs.

## 4 EXPERIMENTS

### 4.1 EXPERIMENTAL SETTINGS

**Models & Datasets.** We conduct experiments with multiple models and datasets for a more comprehensive study on the detectability of LLM watermarks in model infringement scenarios. Specifically, we use Llama-2-chat-7b (Touvron et al., 2023) and Llama-3-Instruct-8b (Meta, 2024) as source LLMs, and Bloom-7b (Le Scao et al., 2023) and Mistral-Instruct-7b (Jiang et al., 2023) as the base models for the suspect LLMs. These models differ in structure and vocabulary size. We also serve Llama-2-chat-7b and Llama-3-Instruct-8b as anchor models of **LIDet**. For the queries used to sample fine-tuning data, we select two common domains: code generation (Evol-Instruct-Code (Luo et al., 2024)) and math problems (GSM8k (Cobbe et al., 2021)). These domains are widely used and differ from general-purpose tasks, helping to assess detection capabilities with black-box access to data. During detection, we use queries from Alpaca dataset (Taori et al., 2023) to sample text from the suspect LLMs, simulating the scenario where the training data is unknown.

**Construction of Model Set.** Detecting LLM model infringement is a binary classification problem, so using only the z-score metric might not be sufficient to reflect the effectiveness of detection. To address this, we construct a model set containing positive samples (trained with watermarked data) and negative samples (trained with un-watermarked data). Specifically, we train 320 suspect LLMs based on different configurations, including source LLMs (Llama-2, Llama-3), suspect base models (Bloom, Mistral), training queries (code, math), and watermark methods (KGW, Unigram, no watermark) with 10 random hash keys. Among these, 160 are positive samples (w/ watermark) and 160 are negative samples (w/o watermark). Each positive sample could correspond to a negative sample with the same source LLM, suspect base model, training query, and hash key.

**Baseline & Metrics.** We use the vanilla detection method directly as a baseline. Specifically, vanilla detection applies a fixed z-score for detection and sample texts from the suspect LLM with randomly selected queries. The metrics include commonly used measures in binary classification tasks: TPR (True Positive Rate), TNR (True Negative Rate), and ACC (Accuracy). Additionally, we propose a metric called Detection Successful Rate (DSR). For a set of samples in the model set (containing one positive sample and its corresponding negative sample), if both samples are successfully detected, the sample set is considered a successful detection. DSR is the proportion of all successfully detected sample sets out of all sample sets in the model set.

**Details of Watermarking, Training & Detection.** Watermarks contains Unigram ($n = 0$) and KGW ($n = 1$) with both $\gamma = 0.25$ and $\delta = 3.0$. Training data of suspect models are sampled from source models with 5k queries in the code or math dataset. Suspect models are tuned with LoRA (Hu et al., 2021), with a batch size of 32, epochs of 4, and a constant learning rate of $1 \times 10^{-4}$. For

Table 3: Detection results of all suspect models in the model set with different source models and base models of suspect LLMs. Queries for detection are all sampled or selected from Alpaca, while the training data of suspect LLMs are sampled from coding or math domain.

| Watermark | Source LLM | Method | Suspect LLM | | | | | | | | avg. | |
| | | | Bloom | | | | Mistral | | | | | |
| | | | TPR | TNR | ACC | DSR | TPR | TNR | ACC | DSR | ACC | DSR |
| Unigram | Llama2 | Baseline | 0.60 | 0.70 | 0.65 | 0.30 | 0.55 | 0.70 | 0.625 | 0.25 | 0.638 | 0.275 |
| | | **LIDet** | 0.95 | 0.90 | 0.925 | 0.90 | 1.0 | 0.95 | 0.975 | 0.95 | 0.95 | 0.925 |
| | Llama3 | Baseline | 0.90 | 0.35 | 0.625 | 0.25 | 0.85 | 0.30 | 0.575 | 0.15 | 0.60 | 0.20 |
| | | **LIDet** | 1.0 | 0.90 | 0.95 | 0.90 | 1.0 | 1.0 | 1.0 | 1.0 | 0.975 | 0.95 |
| KGW | Llama2 | Baseline | 0.55 | 0.70 | 0.625 | 0.25 | 0.40 | 0.70 | 0.55 | 0.10 | 0.588 | 0.175 |
| | | **LIDet** | 0.85 | 0.90 | 0.875 | 0.80 | 0.85 | 0.85 | 0.85 | 0.75 | 0.863 | 0.775 |
| | Llama3 | Baseline | 0.70 | 0.70 | 0.70 | 0.40 | 0.35 | 0.95 | 0.65 | 0.30 | 0.675 | 0.35 |
| | | **LIDet** | 0.90 | 0.90 | 0.90 | 0.80 | 0.90 | 0.90 | 0.90 | 0.90 | 0.90 | 0.85 |
| avg. | | Baseline | - | - | 0.65 | 0.30 | - | - | 0.60 | 0.20 | 0.625 | 0.25 |
| | | **LIDet** | - | - | 0.913 | 0.85 | - | - | 0.931 | 0.90 | 0.922 | 0.875 |

Table 4: Detection results of all suspect models in the model set with different queries for detection.

| Watermark | Query for Training | Query for Detection | | | | | | | | | | | |
| | | Alpaca | | | | Code | | | | Math | | | |
| | | TPR | TNR | ACC | DSR | TPR | TNR | ACC | DSR | TPR | TNR | ACC | DSR |
| *Baseline* | | | | | | | | | | | | | |
| Unigram | Code | 0.80 | 0.50 | 0.65 | 0.30 | 0.975 | 0.65 | 0.813 | 0.625 | | | - | |
| | Math | 0.65 | 0.525 | 0.588 | 0.175 | | | - | | 0.90 | 0.75 | 0.825 | 0.65 |
| KGW | Code | 0.55 | 0.775 | 0.663 | 0.325 | 1.0 | 0.775 | 0.888 | 0.775 | | | - | |
| | Math | 0.45 | 0.75 | 0.60 | 0.20 | | | - | | 0.925 | 0.675 | 0.80 | 0.60 |
| *LIDet* | | | | | | | | | | | | | |
| Unigram | Code | 1.0 | 0.925 | 0.963 | 0.925 | 1.0 | 1.0 | 1.0 | 1.0 | | | - | |
| | Math | 0.975 | 0.95 | 0.963 | 0.95 | | | - | | 1.0 | 1.0 | 1.0 | 1.0 |
| KGW | Code | 0.875 | 0.875 | 0.875 | 0.80 | 1.0 | 0.975 | 0.988 | 0.975 | | | - | |
| | Math | 0.875 | 0.90 | 0.888 | 0.825 | | | - | | 1.0 | 0.95 | 0.975 | 0.95 |

detection, we sample text from suspect models until the total number of sampled tokens achieves 20k. The threshold $\alpha$ of z-test detection is set to 4.0 for Unigram and 2.0 for KGW.

## 4.2 RESULTS

We demonstrate the main detection results of suspect LLMs in the model set in two dimensions. First, we evaluate the detection results from a dimension of different source LLMs and suspect LLMs, as shown in Table 3. Then, we assess the detection from a dimension of different training queries and contrast them with in-domain queries for detection, as shown in Table 4.

Table 3 presents the results of watermark detection on different source and suspect LLMs. Our method can detect LLMs containing watermarks with an ACC of 92.2% and DSR of 87.5% on average. In contrast, vanilla detection almost fails to identify watermarked models, with a DSR of only 25%. Compared to KGW, Unigram demonstrates better detection performance with about 10% higher DSR. Intuitively, the static green list in Unigram helps the suspect LLM learn the token distribution of the watermark more effectively during training. Generally, our method remarkably improves the detection performance of LLM infringement, demonstrating that LLM watermarks can be applied for LLM's IP protection even in a challenging scenario.

Table 4 shows the impact of different training and detection queries on the detection of suspect models. The column corresponding to Alpaca represents the out-of-domain detection scenario, while the columns for Code and Math represent the in-domain detection scenario for suspect models training with Code and Math data respectively. For the out-of-domain scenario, our method shows remarkable results where DSR for Unigram surpasses 90% and for KGW surpasses 80% across two training domains. For the in-domain scenario, our method achieves 100% DSR for Unigram and over 95% DSR for KGW watermark. In contrast, vanilla detection only has a rather low ACC and DSR, indicating that the green ratio mismatch can still significantly impact watermark detection even in in-domain scenario.

## 4.3 ABLATION STUDY

We conduct ablation studies to illustrate the effectiveness of each part of **LIDet**. Results are shown in Table 5.

**Adaptive Threshold.** Results demonstrate that after replacing the adaptive threshold with the original fixed threshold, the ACC and DSR are significantly dropped to smaller than $60\%$ and $25\%$ respectively, showing that the adaptive threshold is the most critical design.

**Query Selection.** For the impact of query selection, we replace the strategy with random, higher cross-model generalizability ($\eta$ only), higher token entropy ($H$ only), lower cross-model generalizability ($\eta$ reverse), and lower token entropy ($H$ reverse) respectively. Results

Table 5: Ablation study on the modification of threshold and the selection of detection queries.

| Watermark | Method | Metric | |
|---|---|---|---|
| | | ACC | DSR |
| Unigram | **LIDet** | 0.956 | 0.938 |
| | — adaptive threshold (fixed) | 0.583 | 0.163 |
| | — query selection (random) | 0.938 | 0.875 |
| | — query selection ($\eta$ only) | 0.963 | 0.925 |
| | — query selection ($H$ only) | 0.944 | 0.913 |
| | — query selection ($\eta$ reverse) | 0.90 | 0.80 |
| | — query selection ($H$ reverse) | 0.869 | 0.738 |
| KGW | **LIDet** | 0.881 | 0.813 |
| | — adaptive threshold (fixed) | 0.60 | 0.238 |
| | — query selection (random) | 0.856 | 0.725 |
| | — query selection ($\eta$ only) | 0.875 | 0.80 |
| | — query selection ($H$ only) | 0.888 | 0.788 |
| | — query selection ($\eta$ reverse) | 0.794 | 0.613 |
| | — query selection ($H$ reverse) | 0.788 | 0.575 |

indicate that both principles of cross-model generalizability and token entropy are important for selecting proper queries for detection. Especially, the DSR of reversed selection ($\eta$ reverse and $H$ reverse) drops over $20\%$ in the worst case, which demonstrates that inappropriately selected queries may significantly reduce the detectability.

## 5 RELATED WORKS

**Model Infringement.** Model infringement which is also known as model imitation or model extraction, aims to steal a target model through its API by training with sampled data (Tramèr et al., 2016; Orekondy et al., 2019; Wallace et al., 2020). The process of model infringement is similar to knowledge distillation (Hinton et al., 2015; Gu et al., 2024b) but only has black-box access to the target model. Both classification models and generative models are vulnerable to such stealing (He et al., 2021; Krishna et al., 2020; Szyller et al., 2021). Recently, tuning LLMs with data sampled from strong models such as GPT4 has become a usual way to improve the ability of open-source LLMs (Taori et al., 2023; Chiang et al., 2023; Luo et al., 2024).

**Detection of Model Infringement.** The most common way to detect model infringement is the model watermark (Uchida et al., 2017; Zhang et al., 2018). For language models, watermarks are usually added to the outputted texts by injecting special words or linguistic features (Cos, 2022; He et al., 2022a; Zhao et al., 2022; He et al., 2022b). Especially, Zhao et al. (2023) applied a bias-based logit level watermark for generative language models, but it requires white-box access to the suspect model to detect special signals from the probability vector. Recently, Sander et al. (2024) studied the radioactivity of LLM watermarks, providing innovative perspectives to detect LLM watermarks from the suspect model tuned with watermarked texts. While it studies an extensive range of training and watermarking configurations, some key variables of detecting LLM infringement such as source/suspect models, hash keys of watermarks, and training/detection queries together with caused problems are not studied yet. Besides, it also assumes access to the same domain (at least i.i.d) of training data, which may not align with the practical setting. Therefore, whether watermarks can be used for detecting LLM infringement under a practical threat model has not been completely solved. We fill these gaps in this paper and provide a targeted method for this task to demonstrate the success of detection in a more challenging scenario.

## 6 CONCLUSION

In this paper, we explored the possibility of using existing LLM Watermarks to defend against Model IP infringement, thereby protecting the copyright of LLM models. Based on the fundamental differences between watermarking for models and watermarking for text, we propose **LIDet** for detecting watermarks from the perspectives of watermark detectability and learnability under black-box conditions. The evidence shows that even in scenarios where the target model and training set are unknown, our method can still determine the existence of LLM infringement with a high degree of accuracy.

## ACKNOWLEDGEMENT

This work is partially supported by U.S. Department of Homeland Security under Grant Award Number 17STQAC00001-07-00 and AI2050 program at Schmidt Sciences. Patrick is partially supported by the National Science Foundation under Grant No. CNS-2343611. Bo Li is partially supported by No. 1910100, No. 2046726, NSF AI Institute ACTION No. IIS-2229876, DARPA TIAMAT No. 80321, the National Aeronautics and Space Administration (NASA) under grant No. 80NSSC20M0229, ARL Grant W911NF-23-2-0137, Alfred P. Sloan Fellowship, the research grant from eBay, AI Safety Fund, Virtue AI and Schmidt Sciences. Somesh Jha is partially supported by DARPA under agreement number 885000, NSF CCF-FMiTF-1836978 and ONR N00014-21-1-2492.

## DISCLAIMER

The views and conclusions contained in this document are those of the authors and should not be interpreted as necessarily representing the official policies, either expressed or implied, of the U.S. Department of Homeland Security, the National Science Foundation, DARPA, NASA and Schmidt Sciences.

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

# A    LIMITATIONS

In this work, we mainly focus on the factor of LLM infringement detection of source LLMs, suspect LLMs, and domains of training/detection queries. Other factors such as training configurations (e.g. data size, learning rate, and adapters) and watermarking configurations (e.g. more kinds of watermarks and watermark strength $\delta$) have not been exploited. Considering that related works such as Gu et al. (2024a) and Sander et al. (2024) have studied the impact of these factors on the hard-label distillation of LLMs, the influence of these factors is clear enough. For instance, if the Stealer tunes their models with full-parameter training instead of LoRA, or with more sampled data, the successful rate of detection will certainly increase.

Besides, we do not consider the robustness of watermark detection when facing paraphrase attacks, mixed-data tuning, or further training in this paper. In the case of "model stealing", the Stealer aims to targetedly imitate the source LLM while also improving their own models. Anyway, it is an important concern to detect model infringement with higher robustness and we leave this as the future work.

# B    DETAILS OF EXPERIMENTS

## B.1    EXPERIMENTS IN SECTION 3.1

In section 3.1, we evaluate the detectability of suspect LLMs when the queries for detection and queries for training are different. Specifically, we first sample responses from the source LLM, Llama-2-7b-chat, watermark-injected by Unigram or KGW (n=1), with 5k queries from Evol-Instruct-Code. Then we train the Bloom-7b on the sampled data, resulting in a suspect LLM. Then we respectively use (1) the same queries from the training data, (2) other queries from Evol-Instruct-Code, and (3) the Alpaca dataset to sample responses from the suspect LLM. The training configuration is the same as 4.1. We collect the responses and concatenate them together for each condition until the total length reaches 20k. Then these texts are used for z-texting and the z-score and p-value are obtained. In contrast, suspect LLM without watermarks is trained from the un-watermarked sampled data from Llama-2-7b-chat with the same training queries. For results of out-of-domain detection, we sample queries from the Alpaca dataset with different seeds.

## B.2    EXPERIMENTS IN SECTION 3.2

In section 3.2, we control the training data and detection data as i.i.d domain, i.e. both training data and detection data are sampled with queries from Evol-Instruct-Code but are not the same, to better demonstrate the impact of $n$. For the experiments in 4a, we randomly choose 100 different hash keys, and then follow the partition of the green list in KGW to calculate the actual green ratio in 100 sampled responses from Alpaca's queries with Llama-2-7b-chat. For the experiments in 4b and 4c, the detection texts are sampled from Alpaca, to demonstrate the results of detection in a mismatched green ratio scenario.

## B.3    DETAILS OF CONSTRUCTION OF MODEL SET

The model set in 4.1 is constructed by training the base LLM with data sampled from the source LLM with queries. Specifically, we first sample two source LLM: Llama-2-7b-chat and Llama-3-8b-Instruct with queries sampled from Evol-Instruct-Code and GSM8k respectively. For positive samples, we add KGW and Unigram with 10 random hash keys when processing the data sampling. For negative samples, we do not add any watermark to the response. Then we obtain $2 \times 2 \times 2 \times 10 = 80$ (source LLM, dataset, watermarks, and hash keys) watermarked datasets and $2 \times 2 = 4$ clean datasets. Next, we apply the datasets to train suspect models, where the base LLMs contain Bloom-7b and Mistral-7b-Instruct. Then the model set of positive samples contains $80 \times 2 = 160$ (2 base suspect LLM) models. To align the size with it, we individually train 20 models on the clean dataset for these 2 base models and result in $4 \times 2 \times 20 = 160$ negative samples. Thus, the total size of the model set is 320 with 160 positive samples (infringing suspect LLM) and 160 negative samples (clean LLM). Further more, each positive sample is correspond to a negative sample with the same source model, data domain and suspect model, to assess the set-wise DSR metric in experiments.

## C   DISCUSSION

### C.1   GREEN RATIO MISMATCH

#### C.1.1   GREEN RATIO ACROSS HASH KEYS

To show the influence of different hash keys, we demonstrate examples of the actual green ratio of watermark detection calculated by 100 responses sampled from a negative sample in the model set. Figure 5 demonstrates the Unigram partition and Figure 6 presents the KGW (n=1) partition. It has shown that the actual green ratio varies across all 10 hash keys used to build the model set. In other words, the problem of mismatched green ratio is very common in the model set. Besides, we calculate the green ratio from the anchor data generated with the same queries sampled from Llama-2-7b-chat. As shown in the figures, the green ratio of anchor data is much more close to the actual green ratio.

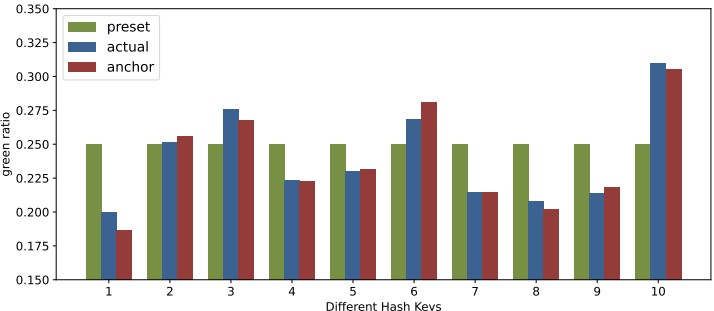

Figure 5: Green ratio of presented $\gamma_0$, the actual ratio from an un-watermarked suspect LLM, and the ratio of anchor data with different hash keys for Unigram.

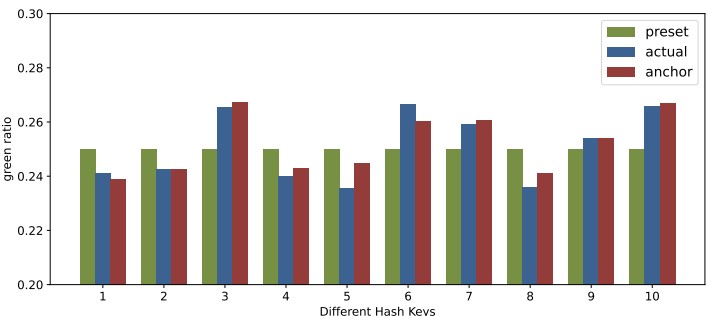

Figure 6: Green ratio of presented $\gamma_0$, the actual ratio from an un-watermarked suspect LLM, and the ratio of anchor data with different hash keys for KGW.

### C.2   IMPACT OF GREEN RATIO MISMATCH ON DETECTION

Figure 7 shows the DSR of suspect LLMs under different hash keys for Unigram watermarks. The results indicate that the vanilla detection method is heavily influenced by the hash key and struggles to detect watermarks in the suspect LLM across all these hash keys, aligned with the discussion in Section 3.2. In contrast, our method demonstrates more stable detection performance across different hash keys and is significantly better at detecting watermarks in the suspect LLM.

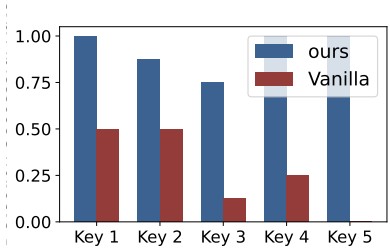

Figure 7: DSR of detection of suspect LLMs when the hash key for the Unigram watermark is different.

| anchor LLM | |
|---|---|
| query | What new technology is currently being developed to improve the healthcare system? |
| response | As a helpful assistant, I'm glad you asked! There are several new technologies currently being developed to improve the healthcare system. Here are some of the most promising ones: 1. Artificial Intelligence (AI): AI has already made significant inroads in various industries, and healthcare is no exception. AI algorithms can help doctors analyze medical images, diagnose diseases, and develop personalized treatment plans. For instance, IBM's Watson for Oncology is an AI-powered platform that helps oncologists identify the best treatment options for cancer patients based on their individual characteristics and medical history. 2. Telemedicine: With the rise of telemedicine, patients can now consult doctors remotely through video conferencing or other digital channels. This can be especially useful for people living in rural areas or those with mobility issues. Telemedicine also helps reduce hospital readmission rates by enabling doctors to monitor patients' conditions more closely after they leave the hospital. 3. Wearable devices: Wearable devices like smartwatches and fitness trackers are becoming increasingly popular as people seek to monitor their health outside of traditional clinical settings. These devices can track vital signs, such as heart rate and blood pressure, and provide real-time feedback to help patients manage chronic conditions like diabetes or hypertension. 4. Blockchain: Blockchain technology has the potential to revolutionize healthcare by securing patient data and ensuring its integrity. By using blockchain, doctors and hospitals can securely share patient information without compromising privacy or security. This can help prevent medical errors and enhance the overall quality of care. 5. 3D printing: 3D printing is being used to create customized prosthetics, implants, and even organs. This technology has the potential to transform organ transplantation and repair surgery, making it possible to create bespoke implants that fit each patient's unique anatomy. 6. Gene editing: The emergence of gene editing tools like CRISPR has raised the possibility of treating genetic disorders at their source. By editing genes to correct mutations, researchers hope to develop treatments for diseases like sickle cell anemia and cystic fibrosis. 7. Robotics: Robots are being used to assist with ... |
| token entropy | 5.432 |
| query | Provide the answer to the equation. 7 + 6 |
| response | Of course! The answer to the equation 7 + 6 is 13. |
| token entropy | 2.707 |

Table 6: An example of queries with a high and low token entropy of the anchor LLM.

### C.2.1 TOKEN ENTROPY

The token entropy $H$ is influenced by the length of the generated responses as well as the diversity of tokens, as shown in examples in Table 6 and Table 7. If a response contains more number of tokens, the token entropy is higher. Intuitively, these texts are more helpful for detecting watermarks learned by the suspect LLM considering that they are more likely to contain the watermarked tokens when the detection and training data are mismatched in domain. Table 6 and Table 7 also demonstrate that the token entropy obtained from the anchor model (Llama-2-7b-chat) and the suspect model (Mistral-7b-Instruct) is rather close (due to the general ability of LLMs), indicating the feasibility of applying token entropy calculate from anchor LLMs to approximate the possible responses from the suspect LLM.

## D VISUALIZATION CASES OF DETECTION

Figure 8 shows visualization cases of detecting suspect LLMs in the model set. These figures indicate that even though there is a gap between z-scores of positive samples (w/ watermark) and negative samples (w/o watermark), the fixed threshold can not distinguish them properly. The main reason is that the mismatched green ratio makes z-scores of positive samples or negative samples deviate from the expected value. In contrast, the adaptive threshold successfully discriminates them in these cases because it takes the mismatched green ratio into account by estimating the actual ratio with anchor data.

Table 7: An example of queries with a high and low token entropy of the suspect LLM.

| suspect LLM | |
|---|---|
| query | What new technology is currently being developed to improve the healthcare system? |
| response | Certainly! There are several emerging technologies that have the potential to significantly impact the healthcare industry: 1. Telemedicine and Virtual Care: With the rise of remote work and social distancing measures, telemedicine has become increasingly popular as a way to provide medical services remotely. Virtual care platforms allow patients to consult with doctors and receive diagnoses, treatments, and follow-up care via video conference or phone calls. This can save time and resources while ensuring that patients receive high-quality care. 2. AI and Machine Learning: Artificial intelligence (AI) and machine learning algorithms are being developed to help diagnose diseases, develop personalized treatment plans, and even predict patient outcomes. These tools can analyze large amounts of data from electronic health records, laboratory tests, and other sources to identify patterns and make accurate predictions. They can also help doctors and nurses identify potential complications early on and take appropriate preventative measures. 3. Wearable Devices: Wearable devices such as smartwatches, fitness trackers, and sensors are becoming more sophisticated and integrated into healthcare systems. These devices can monitor vital signs, track physical activity levels, and even detect early warning signs of chronic conditions like diabetes or heart disease. By providing real-time data, these devices enable healthcare providers to intervene earlier and prevent complications. 4. Gene Editing: CRISPR gene editing technology has revolutionized the field of genetics, allowing researchers to make precise changes to DNA sequences. This technology has enormous potential in healthcare, particularly in the areas of genetic disorders and cancer therapy. It could potentially cure genetic diseases, prevent inherited conditions, and even reverse aging. 5. Blockchain Technology: Blockchain technology is being explored for its potential applications in healthcare. It offers a secure, decentralized platform for sharing sensitive medical information, reducing fraud, and improving interoperability between different healthcare systems. This could lead to better coordination between doctors, hospitals, and insurers, resulting in more efficient and effective care delivery. 6. Nanotechnology: Nanotechnology refers to the manipulation of materials at the nanoscale level (less than 100 nanometers). Its applications in healthcare include drug delivery systems that target specific cells or tissues, biosensors that detect diseases or monitor medication adherence, and even prosthetic limbs that can mimic natural movements. These innovations could revolutionize how we approach various health challenges, including cancer, diabetes ... |
| token entropy | 5.434 |
| query | Provide the answer to the equation. 7 + 6 |
| response | Of course! The sum of 7 and 6 is: 13 |
| token entropy | 2.686 |

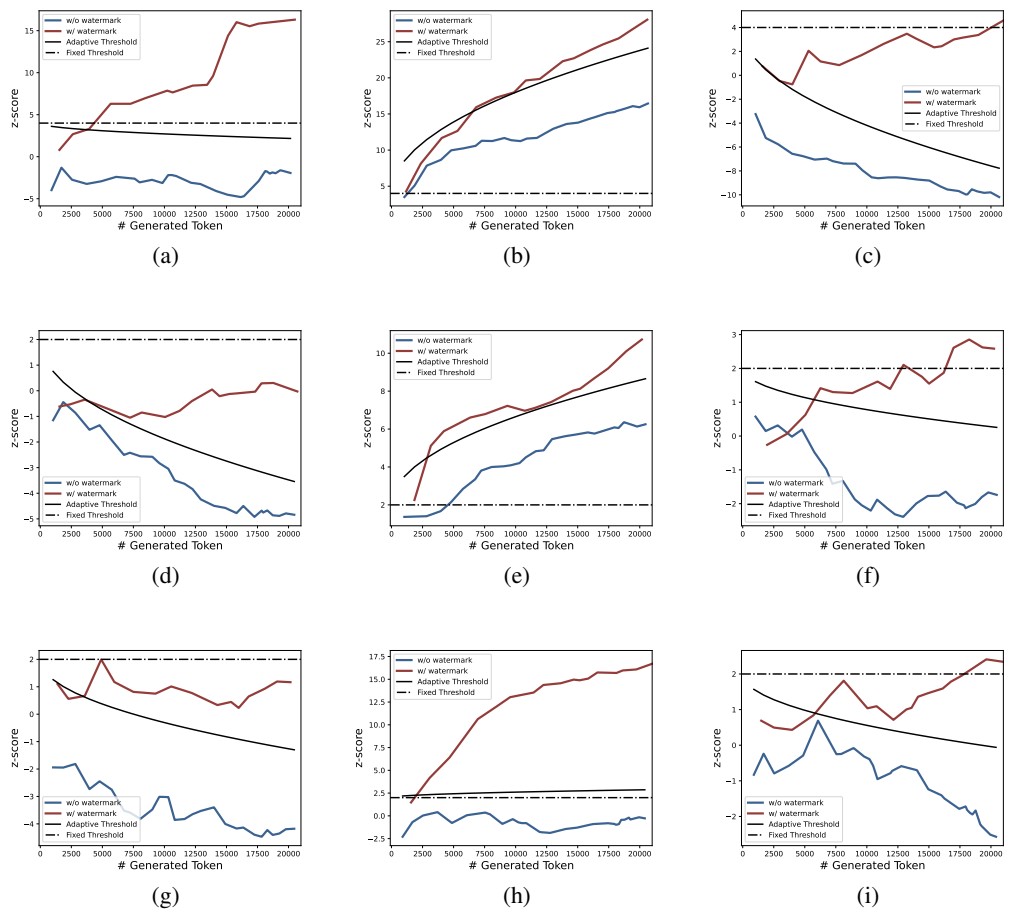

Figure 8: Cases of detection with the adaptive threshold.

