# OpenReview forum: "Can Watermarks be Used to Detect LLM IP Infringement For Free?"
_ICLR.cc/2025/Conference — ICLR 2025 Poster_

### Official Review · Reviewer_z7oZ · 2024-10-31

**Soundness:** 3
**Presentation:** 3
**Contribution:** 3
**Rating:** 6
**Confidence:** 2

**Summary:**

This paper proposes using watermarking to detect IP infringement in large language models (LLMs). The proposed method, LIDet, improves detection accuracy by addressing domain mismatch and watermark stability through adaptive thresholding and optimized query selection based on token entropy and generalizability. Experiments show that LIDet achieves high reliability in distinguishing between infringing and non-infringing models, enhancing LLM protection.

**Strengths:**

The proposed approach in this paper leverages watermarking as a novel method to detect IP infringement in large language models (LLMs), providing a robust layer of protection against unauthorized replication. The proposed detection method, LIDet, tackles key challenges such as domain mismatch and stability issues through adaptive thresholding and a targeted query selection strategy. By focusing on token entropy and cross-model generalizability, the proposed query strategy ensures more reliable detection, reducing false positives and achieving high accuracy across various settings.

**Weaknesses:**

The paper could benefit from further exploration of the impact of watermarking on overall model performance, including metrics such as perplexity, BLEU scores, or human evaluations to assess any trade-offs in output quality and diversity. Additionally, while candidate queries are selected for high token entropy and cross-model generalizability to ensure that infringing models retain watermark patterns, the detection reliability may vary if infringing models do not fully absorb specific watermarked phrases. Testing detection on models trained with varying amounts of watermarked data could clarify this absorption. Lastly, the “anchor LLM” method aims to reduce false positives by selecting queries with favorable green ratios and entropy, but there is still a risk that unwatermarked models could unintentionally exceed detection thresholds. Analyzing the false positive rate across a broader set of non-infringing models under various conditions would provide a more comprehensive understanding of the method’s specificity.

**Questions:**

1. If two models are trained on overlapping open-source datasets, will the false positive rate significantly increase? Shared data could lead to misclassification of non-infringing models as infringing ones.
2. How does watermarking affect the model’s performance—does it enhance or degrade output quality? It’s essential to evaluate whether watermarks could unintentionally reduce the overall effectiveness of the model's generation capabilities.
3. How can candidate queries effectively ensure that the infringing model has learned specific watermark phrases? If the infringing model has not learned the intended watermarked phrases, can these queries reliably retrieve them?
4. How can we ensure that the responses retrieved through queries are indeed related to the watermark? Some models may unintentionally produce watermarked phrases simply through regular training, which could lead to inadvertent misclassifications.

---

> ### Author Response · Authors · 2024-11-24
> **Thanks for your invaluable comments and advice. (1/2)**
>
> #### **Question 1: Considerarion of Overlapping Dataset**
>
> The false positive rate of the detection would not be influenced by the overlapping unwatermarked data of two models. The detection of infringing models relies on identifying the watermark, which increases the proportion of "green" tokens and reduces the proportion of "red" tokens, as shown in [1]. The accuracy of detection is theorically ensured through a z-test, with the error rate being extremely low (0.003%) when the z-score exceeds 4. This means that the overlapping data will not contribute to the watermark as long as its green token ratio does not exceed a rather high threshold which is amost impossible to reach for unwatermarked text. Therefore, even if two models are trained on the same dataset without a watermark, a non-infringing model will not exhibit the green ratio shift induced by a specific watermark, making it highly probable that the model is not infringing.
>
> #### **Question 2 & Weakness 1: Impact on General Performance**
>
> The impact of LLM watermarking on model performance is an important issue and has been widely studied. In the watermarking method we selected, Unigram [2] has been shown in its experiments to maintain high effectiveness with minimal impact on general performance. To further demonstrate the effect of fine-tuning on suspect LLMs using watermarked training data, we evaluated the perplexity of different models. Specifically, we used Alpaca-7b to calculate perplexity, and the results are shown in the following table.
>
> | Metric | Source Model | Source Model | Suspect Model | Suspect Model |
> | --- | --- | --- | --- | --- |
> | Model | Llama2 | Llama2 | Mistral | Mistral |
> | Watermark | w/o watermark | w/ watermark | w/o watermark | w/ watermark |
> | Avg. PPL | 3.96 | 5.09 | 4.17 | 4.54 |
>
>
> Our results are similar to those presented in [2], showing that watermarking has a minimal effect on general performance. However, this impact is controllable, and by selecting watermarking methods like Unigram, which have a minimal impact on general performance, the performance degradation caused by watermarking can be effectively reduced. Furthermore, compared with the impact of watermarks on directly generated text, the results demonstrate that the impact on distilled suspect model is less significant.
>
> #### **Question 3 & Weakness 2: Learnability of Watermarks & Amounts of Watermarked Data.**
>
> Given that watermark detection is related to the proportion of green tokens, watermarked data will have a green token ratio that deviates from the mean. After fine-tuning on watermarked data, the proportion of these green tokens in the suspect model's output will change. If the query used for detection contains these green tokens, the green ratio shift can be detected. However, domain mismatch between the training data and the query may lead to detection failure. Therefore, our method selects queries with high token entropy to maximize token diversity, increasing the likelihood of detecting green tokens.
>
> We also evaluate the detectability of the suspect model with varying amounts of watermarked data. Specifically, we set the sizes of the watermarked training data to 1000, 2000, 3000, 4000, and 5000, using the Unigram watermark, with the source model as Llama2 and the suspect base model as Mistral. The experimental results are shown in the table below.
>
> | Method | datasize | TPR | TNR | ACC | DSR |
> | --- | --- | --- | --- | --- | --- |
> | LIDet | 1000 | 1.0 | 0.9 | 0.95 | 0.9 |
> |  | 2000 | 1.0 | 0.9 | 0.95 | 0.9 |
> |  | 3000 | 1.0 | 0.9 | 0.95 | 0.9 |
> |  | 4000 | 1.0 | 0.9 | 0.95 | 0.9 |
> |  |
> | Baseline | 1000 | 0.5 | 0.7 | 0.6 | 0.2 |
> |  | 2000 | 0.5 | 0.7 | 0.6 | 0.2 |
> |  | 3000 | 0.6 | 0.7 | 0.65 | 0.3 |
> |  | 4000 | 0.7 | 0.7 | 0.7 | 0.4 |
>
>
> The results show that our method can effectively detect infringing models, even when only a small amount of watermarked data is used for training.

---

> ### Author Response · Authors · 2024-11-24
> **Thanks for your invaluable comments and advice. (2/2)**
>
> #### **Question 4 & Weakness 3: Risk on Unwatermarked Models.**
>
> By conducting a z-test on the proportion of green tokens in the text, we can determine how much the green ratio deviates from the set threshold for green tokens. This ensures that unwatermarked models are classified correctly with high probability. Even if these models produce watermarked phrases (green tokens), as long as the green token ratio's z-score does not exceed the threshold, they will not be classified as infringing models. (i.e. For an unwatermarked text, the probability of its z-score exceeding 2.0 is less than 0.023, and the probability of exceeding 4.0 is less than 0.00003.)
>
> However, in practice, there is often a mismatch between the set green ratio and the actual green ratio in natural, unwatermarked text, which increases the FPR during detection. To mitigate the impact of this green ratio mismatch, we introduce anchor data to adjust the green ratio of unwatermarked text, thereby reducing FPR and improving overall detection success.
>
> We demonstrate the FPR of unwatermarked models with the detection of the Unigram watermark, showing the accuracy of our method in detecting unwatermarked models.
>
> | Unwatermarked Models | Llama2 | Llama3 | Bloom | Mistral | Vicuna-7b | Guanaco-7b | Alpaca-7b |
> | --- | --- | --- | --- | --- | --- | --- | --- |
> | FPR | 0.00 | 0.05 | 0.1 | 0.03 | 0.03 | 0.05 | 0.00 |
>
>
>
> [1] Kirchenbauer et al, A watermark for large language models, ICML 2023.
>
> [2] Zhao et al, Provable Robust Watermarking for AI-Generated Text, ICLR 2023.

---

> > ### Comment · Reviewer_z7oZ · 2024-11-25
> > **Reviewer Feedback**
> >
> > Thank you. The author has addressed all the concerns I raised.

---

### Official Review · Reviewer_azDs · 2024-11-02

**Soundness:** 3
**Presentation:** 3
**Contribution:** 3
**Rating:** 6
**Confidence:** 3

**Summary:**

This paper investigates using watermarks in large language models (LLMs) to detect intellectual property (IP) infringement. It identifies two key issues with direct watermark detection: the impact of sampling queries on detectability and the instability of watermark hash keys. To address these, the authors propose LIDet, a method that selects suitable sampling queries and adapts detection thresholds. Experimental results show LIDet achieves over 90% accuracy in distinguishing between infringing and clean models, demonstrating its effectiveness for detecting LLM IP infringement.

**Strengths:**

1. The paper is well-written, with clear logic and organization that makes it easy to follow.
2. The experimental results appear promising, demonstrating the proposed method's effectiveness in distinguishing between infringing and clean models.

**Weaknesses:**

1. Evaluation of Time/Computational Costs: The experimental section lacks an assessment of the time and computational costs involved. It should evaluate the number of queries required for effective detection, as this is crucial for understanding the practicality of the proposed method.
2. Transferability of the Approach: The paper does not adequately address how the proposed method, LIDet, can be applied to other models. Clarifying the conditions under which the method retains its effectiveness across different architectures would strengthen the contribution of the work.

**Questions:**

(1) Please evaluate the number of queries required for effective detection and present the results obtained using varying numbers of queries.

(2) It is recommended to conduct experiments using models other than Mistral-Instruct-7b as the base model.

---

> ### Author Response · Authors · 2024-11-24
> **Thanks for your invaluable comments and advice.**
>
> #### **Weakness 1: Computational Cost.**
>
> We demonstrate the detection accuracy of our method under different numbers of queries and response tokens. In the case of the Unigram watermark, with the domain being Code, source model Llama2, and suspect base model Mistral, we measure detectability under varying output token counts. The experimental results are shown in the table below.
>
> | Method | #token | Avg. #query | TPR | TNR | ACC | DSR |
> | --- | --- | --- | --- | --- | --- | --- |
> | LIDet | 2000 | 4.2 | 0.7 | 1.0 | 0.85 | 0.7 |
> |  | 4000 | 8.2 | 0.9 | 0.9 | 0.9 | 0.8 |
> |  | 6000 | 12.4 | 1.0 | 0.9 | 0.95 | 0.9 |
> |  | 8000 | 16.5 | 1.0 | 0.9 | 0.95 | 0.9 |
> |  | 10000 | 20.8 | 1.0 | 0.9 | 0.95 | 0.9 |
> |  |
> | Baseline | 2000 | 10.2 | 0.4 | 0.8 | 0.6 | 0.2 |
> |  | 4000 | 17.5 | 0.5 | 0.8 | 0.65 | 0.3 |
> |  | 6000 | 26.2 | 0.5 | 0.8 | 0.65 | 0.3 |
> |  | 8000 | 35.1 | 0.5 | 0.8 | 0.65 | 0.3 |
> |  | 10000 | 45.8 | 0.6 | 0.7 | 0.65 | 0.3 |
>
> The experimental results show that our method achieves a high detection rate even with a smaller number of detection tokens. This is due to the adaptive adjustment of the z-score threshold, which allows effective differentiation of positive and negative samples even with fewer tokens, as shown in Figure 8. Increasing the number of detection tokens further enhances the z-score gap between positive and negative samples, thereby increasing the confidence in detection.
>
> #### **Weakness 2: Transferability.**
>
>
> In our experiments, we select four different models with varying structures as source or suspect base models. The source models include Llama2 and Llama3, while the suspect models include Mistral and Bloom. This demonstrates that our method works effectively across models with different structures. To further showcase its effectiveness, we conducted new experiments using Mistral as the source model and Llama2 as the suspect base model. The experimental results are presented in the table below.
>
> | Method | Source LLM | Suspect LLM | TPR | TNR | ACC | DSR |
> | --- | --- | --- | --- | --- | --- | --- |
> | LIDet | Mistral | Llama2 | 1.0 | 0.9 | 0.95 | 0.9 |
> | Baseline | Mistral | Llama2 | 0.9 | 0.6 | 0.75 | 0.5 |
>
>
> The experiments show that regardless of the structural differences between the source and suspect models, our method can effectively detect infringing suspect models with a high probability.

---

### Official Review · Reviewer_aqr8 · 2024-11-03

**Soundness:** 2
**Presentation:** 2
**Contribution:** 3
**Rating:** 5
**Confidence:** 3

**Summary:**

To distinguish the AI-generated texts from human-produced ones, watermarking is added over generated texts, by using green-and-red list and adding bias to the logits of different tokens. In this paper, the authors exploit this text watermark to detect whether a new model $M_a$ is finetuned over the texts generated from a well-trained model $M_b$. Ideally, if $M_a$ is totally trained over the texts generated from $M_b$, the outputs of $M_a$ satisfy the same bias as $M_b$, so that the detector can detect the output of $M_a$ to determine whether the IP of $M_b$ is violated or not. However, in the realistic situation, there exist the domain mismatch and green ratio mismatch between $M_a$ and $M_b$, which leads to the failure of the detection. The authors proposed a new method called LIDet to detect whether $M_a$ is finetuned over the $M_b$-generated data.

**Strengths:**

1.	Clear definition of threat model to show the capability of stealer and detector.
2.	Scrutiny the reasons on the attenuation of watermark detection: domain mismatch and green ratio mismatch.

**Weaknesses:**

Lack of adaptive attack: The authors only assume that the stealer will directly finetune $M_a$ over the queried data, and fail to exploit the adaptive attack. For instance, if the stealer knows the queried data including the watermark, he/she could modify the queried texts to remove the watermark. By this way, the finetuned model $M_a$ may not learn the bias. From another perspective, the stealer can even finetune the $M_a$ without the data from $M_b$, but mimic the same bias as $M_b$ to increase the FPR of LIDet.

**Questions:**

NA

---

> ### Author Response · Authors · 2024-11-24
> **Thanks for your invaluable comments and advice.**
>
> Thank you for your suggestion. We consider the adaptive attack from the following considerations.
>
> #### **Consideration of Watermark Removal.**
> The robustness of LLM watermarkings towards watermark removal attacks has been carefully studied by previous watermarking methods. In the watermarking method we selected, Unigram [1] has been theoretically proven to be more resistant than some other watermarking methods to attacks like paraphrasing, which attempt to remove the watermark through post-processing. We perform a paraphrasing attack on Code data sampled from the source model Llama2 using Mistral and fine-tune the suspect model on the paraphrased data. The experimental results, shown in the table, indicate that while paraphrasing weakens the detectability of the watermark, it does not completely prevent the suspect model from being detected.
>
> | Method | Attack | TPR | TNR | ACC | DSR |
> | --- | --- | --- | --- | --- | --- |
> | LIDet | No Attack | 1.0 | 0.9 | 0.95 | 0.9 |
> |  | Paraphrase | 0.6 | 0.9 | 0.75 | 0.5 |
>
> In practical scenarios, a stealer often requires high-quality training data. Using methods like paraphrasing to alter data from the source model could reduce the quality. Specifically, if a model used for modification or paraphrasing is strong enough to maintain the quality of the original data, the stealer might prefer directly generating training data with it rather than sampling from the source model. As a result, our threat model does not consider modifications such as paraphrasing made by the stealer to the training data.
>
> #### **Consideration of Watermark Mimicry.**
>
> While a stealer could attempt to mimic the same bias to increase the false positive rate (FPR) in detection, this would incur a significant cost with limited benefit in the scenario of LLM IP infringement. Firstly, since LLM watermarking subtly adjusts the logits during fine-tuning rather than explicitly injecting special characters into the text, it is less visible. Mimicking this watermark bias would require extensive statistical effort. Furthermore, doing so would force stealers to provide more evidence when they face infringement accusations from providers of the source model, which will create more inconvenience, diminishing their motivation to mimic the watermark bias.
>
> [1] Zhao et al, Provable Robust Watermarking for AI-Generated Text, ICLR 2023.

---

> > ### Comment · Reviewer_aqr8 · 2024-11-26
> >
> > Thanks for your responses. My concern about watermark removal is clarified, but not the mimicry.
> >
> > I understand that adding a watermark will lead to a tiny part of performance degradation, but if the stealer can generate another model fine-tuned over non-queried data (from the victim model) and meanwhile mimic with watermark bias. It will lead to a high FPR of the infringement detector, which causes the failure of the infringement detector. My question is whether a stealer can successfully generate this kind of mimic model. Thanks.

---

> > > ### Author Response · Authors · 2024-12-02
> > > **Request for Your Invaluable Feedback**
> > >
> > > Dear Reviewer aqr8,
> > >
> > > Thanks again for your positive response to our rebuttal and we have explained the feasibility of the mimicry attack in our recent response. We sincerely request your further feedback to let us know whether your concern has been addressed.
> > >
> > > Best regards, Authors

---

> ### Author Response · Authors · 2024-11-26
> **Response by Authors**
>
> Thank you for your feedback.
>
> Executing a mimicry attack requires the stealer to replicate the bias introduced by the watermark. The bias in an LLM watermark depends on a hash key owned by its provider, which remains confidential to the stealer. Without access to this key, one potential approach involves extensive sampling of the target LLM to analyze token or N-gram proportions, thereby inferring the green list in the watermark configuration. However, this process is highly resource-intensive. For instance, [1] demonstrates that successfully mimicking an unknown watermark, such as KGW, requires 1 million queries to the target LLM. This query volume far exceeds the size of data typically used for instruction fine-tuning, making it an impractical and costly endeavor for the stealer.
>
> A more efficient method to mimic such bias is using data sampled from the target LLM to fine-tune another LLM (serving as an imitation model) [2,3]. However, in our threat model, leveraging data from the source LLM to fine-tune an imitation model also constitutes infringement. In summary, if the stealer aims to generate another model fine-tuned over non-queried data while mimicking watermark bias, they must either extensively sample the victim model (incurring prohibitive costs) or train an imitation model (considered infringement and correctly classified by our detector).
>
> [1] Sadasivan, et al. Can AI-Generated Text be Reliably Detected? arxiv, 2023.
>
> [2] Gu, et al. On the Learnability of Watermarks for Language Models, ICLR 2024.
>
> [3] Jovanovic, et al. Watermark Stealing in Large Language Models, ICML 2024.

---

### Official Review · Reviewer_gu9D · 2024-11-04

**Soundness:** 2
**Presentation:** 3
**Contribution:** 2
**Rating:** 6
**Confidence:** 2

**Summary:**

This paper studies the problem of Model IP infringement for LLMs. The authors propose to use different anchor LLMs to select query samples that are more likely to contain more diverse and evenly distributed tokens, which are useful for later detection. Furthermore, for the detection phase, it uses dynamic thresholding for the z-score-based detection methods. The empirical results show the significant detection performance compared to the baselines.

**Strengths:**

1. The structure of the paper is good that, it first empirically identifies the vulnerabilities of existing watermarking scheme for text or models.
2. The empirical performance is strong.

**Weaknesses:**

1. The evaluation settings rather limited and only contains two domains and also did not consider the fact that, if the target domain is a mixture of multiple domains (e.g., math and code). The paper can benefit from a more comprehensive evaluations that mixes multiple domains.
2. The threat model is still unclear to me. Even though the different types of downstream applications exist, the the source model trainer should still have some general understanding about the high-level domains that are involved in the training set. Therefore, even for the vanilla approach, generating some possible responses from each of these high-level domains is a possible (and potentially stronger) baseline.

**Questions:**

As the paper is mostly an empirical paper, I would suggest making the evaluations more comprehensive. The detailed suggestions for comprehensive evaluations are listed in the weakness section.

---

> ### Author Response · Authors · 2024-11-24
> **Thanks for your invaluable comments and advice.**
>
> #### **Weakness 1: Results of Mixed Domain.**
>
> In our experiment, we consider two domains, Code and Math, because they are common applications in LLM instruction tuning, and many models are fine-tuned specifically for code/math tasks (e.g., CodeLlama). Additionally, the data obtained from Code and Math often differ significantly from general task data, posing a greater challenge for detecting out-of-domain data in our threat model when the suspect model's training data is unknown. When training data from different domains is mixed, the distribution becomes broader, increasing the variety of tokens affected by the watermark, which aids in watermark detection in the suspect model. Therefore, we mix Code and Math in a 1:1 ratio using the Unigram watermark, with the source model as llama2 and the suspect model as mistral. The resulting training data contains 5000 query-response pairs (2500 code and 2500 math), and we fine-tune the suspect model under this setup. The detection results for the suspect model are as follows.
>
> | Method | Domain | TPR | TNR | ACC | DSR |
> | --- | --- | --- | --- | --- | --- |
> | LIDet | Code+Math | 1.0 | 0.9 | 0.95 | 0.9 |
> | Baseline | Code+Math | 0.7 | 0.7 | 0.7 | 0.4 |
>
> #### **Weakness 2: Consideration of High-level Domains.**
>
>
> Detecting the suspect model by constructing queries from various high-level domains is a possible method. However, high-level domains are difficult to define, and
> the detector faces challenges in selecting a set of queries that cover all potential domains. This approach can also incur high costs. For example, domains with minimal token overlap could result in ineffective queries during detection, as certain domain-specific queries might not contribute to the watermark detection process. In contrast, we approach watermark detection from the token perspective rather than the domain perspective. By covering a wide range of tokens in general queries, we can more efficiently detect watermarks without prior knowledge of the target domain. This method reduces reliance on specific domains and enhances the detection process.
>
> Furthermore, our observations show that even when the detector fully knows the target domain, the detectability of the suspect model remains low due to green ratio mismatches in LLM watermarks like Unigram (Table 4). Therefore, our two-stage method mitigates both domain mismatch and green ratio mismatch, significantly improving watermark detectability with minimal cost.

---

> > ### Comment · Reviewer_gu9D · 2024-11-25
> >
> > Thanks for the rebuttal. My concerns are addressed and I have updated my score accordingly!

---

### Official Review · Reviewer_RZ9s · 2024-11-04

**Soundness:** 3
**Presentation:** 3
**Contribution:** 2
**Rating:** 6
**Confidence:** 3

**Summary:**

The authors propose LIDet, a novel method designed to detect whether an LLM has been fine-tuned using the outputs of another LLM, potentially causing IP infringement. The approach assumes that the source LLM’s generated outputs are watermarked and that a suspect model is fine-tuned on them.  Based on this and assuming that the suspect model has learned this watermarked distribution of data, the method uses a set of anchor LLMs to select appropriate queries for sampling from the suspect model. It then detects watermarks in the suspect's generated text using an adaptive threshold based on z-scores. The authors evaluate the method with different models and datasets, proposing an analysis and ablation study of LIDet based on different watermarking methods.

**Strengths:**

•	The authors propose an evaluation of the method with data from different domains.

•	The authors evaluate the performance of the detection method with different watermarking techniques.

•	The authors propose a clear and precise problem statement and threat model.

**Weaknesses:**

•	 The paper lacks in the comparison of LIDet with other methods in the state of the art.

•	The Results section (4.2) does not provide sufficient details on the performance of the different anchor models considered in LIDet.

•	The authors do not propose an evaluation of the method using different combinations of original and suspect models.


**Comments.**

1.	In the related works, the authors cite the work in [1] which proposes another method to detect LLM watermarks from the suspect model tuned with watermarked texts. The authors did not propose a comparison of LIDet with this method, even though it attempts to solve the same type of problem. Although the threat model is slightly different, the author could have proposed a comparison with [1] in Table 3 when the Query for Detection follows the same distribution as the Query for Training, i.e., i.i.d. detection queries as considered in [1].

2.	In the first step of the detection method, the authors propose to sample a set anchor LLMs $\theta_{\rm anchor}^M$ used to help select the proper queries to sample the suspect LLM. As presented in the paper, the set of anchor models presents the same model architectures as the source model, however, there is no discussion on how to select the right set of anchor models, nor is there any insight into which anchor model contributes the most to the selection of the final set of queries. The paper does not present any ablation on the number of anchor models or on the type of model architectures.

3.	In addition to that, the authors considered only the scenario where Llama2 and Llama3 models are used as source and anchor models, but not as suspect models. There is no analysis of LIDet's performance when the suspect and source models are derived from the same base model. Similarly, the authors did not provide any analysis of the method using different model architectures as source models, e.g. Mistral.



**References**

[1] Tom Sander, Pierre Fernandez, Alain Durmus, Matthijs Douze and Teddy Furon, Watermarking Makes Language Models Radioactive, 2024

**Questions:**

•	Do you think LIDet performs better than [1]? Could you provide a comparison of the two methods?

•	Should the set of anchor models always contain the same source model architecture?

•	Do you think the suspect model's dimension (number of parameters) could affect the performance of the method?

•	Do you think there is a correlation between the detectability of the watermarks and the domain of the datasets considered in the queries used to train the suspect model?

---

> ### Author Response · Authors · 2024-11-24
> **Thanks for your invaluable comments and advice. (1/2)**
>
> #### **Weakness 1: Comparison with Baseline [1] (i.i.d condition).**
>
> Our threat model assumes that the detector has no knowledge of the stealer's training data distribution. In this scenario, the method proposed in [1], which relies on constructing i.i.d. watermarked data to calculate filters during detection, is not directly applicable. To facilitate comparison with [1], we hypothesize that the detector knows the stealer's training data domain (e.g., Code or Math) and implements their approach under this assumption. Specifically: (1) For Unigram: Since its green list does not depend on previous tokens, we only perform de-duplication of [1] on the detection data. (2) For KGW (k=1): We first generate reference watermarked data using the corresponding source model. Based on this reference data, we construct filters and apply de-duplication during detection.
>
> During testing, 20,000 tokens are sampled for each model using prompts tailored to the domain (Code or Math). Experimental results are summarized in the table below.
>
> | Method | Watermark | Domain | TPR | TNR | ACC | DSR |
> | --- | --- | --- | --- | --- | --- | --- |
> | [1] | Unigram | Code | 0.825 | 0.75 | 0.7875 | 0.575 |
> |   |   | Math | 0.725 | 0.725 | 0.725 | 0.45 |
> |   | KGW | Code | 1.0 | 1.0 | 1.0 | 1.0 |
> |   | | Math | 0.725 | 1.0 | 0.8625 | 0.725 |
> |  |
> | LIDet | Unigram | Code | 1.0 | 1.0 | 1.0 | 1.0 |
> |   |   | Math | 1.0 | 1.0 | 1.0 | 1.0 |
> |   | KGW | Code | 1.0 | 0.975 | 0.988 | 0.975 |
> |   | | Math | 1.0 | 0.95 | 0.975 | 0.95 |
>
>
> Experimental results demonstrate that when the training data domain of the suspect model is known, [1] can effectively detect infringing models with KGW watermarks. However, its performance is suboptimal with Unigram due to greater green ratio variability caused by different hash key divisions (Figure 4a), which de-duplication fails to mitigate. In contrast, our method exhibits more stable performance under i.i.d. conditions. By introducing a correction term for the green ratio using anchor data, our approach significantly reduces detection failures caused by green ratio mismatches.
>
>
> #### **Weakness 2: Impact of Anchor Models.**
>
> In our method, the Anchor Model is used to approximate the green token ratio in natural text (without watermark) under different green list partitions caused by hash keys, thereby addressing the green ratio mismatch issue. Additionally, it helps filter queries that elicit richer token diversity in the suspect model's responses. The requirement for the Anchor Model is generality, reflecting the natural text distribution. Experiments using various models with different structures and scales as anchor models (Unigram watermark, Code data) demonstrate the feasibility of leveraging versatile LLMs for this purpose. Results are presented below.
>
> | Anchor Model | TPR | TNR | ACC | DSR |
> | --- | --- | --- | --- | --- |
> | Llama2 | 1.0 | 0.95 | 0.975 | 0.95 |
> | Llama3 | 1.0 | 0.9 | 0.95 | 0.9 |
> | Bloom | 1.0 | 0.9 | 0.95 | 0.9 |
> | Mistral | 1.0 | 1.0 | 1.0 | 1.0 |
> | Llama2+Llama3 | 1.0 | 0.9 | 0.95 | 0.9 |
> | Llama2+Llama3+Bloom | 1.0 | 0.9 | 0.95 | 0.9 |
> | Llama2+Llama3+Bloom+Mistral | 1.0 | 0.95 | 0.975 | 0.95 |
>
> Experimental results indicate that using anchor models with varying architectures and scales consistently achieves high detection accuracy. This demonstrates that despite differences in model designs, these LLMs exhibit alignment in their general data distributions. Notably, when the anchor model matches the base model of the suspect LLM, detection accuracy improves (e.g. Mistral). This suggests that the detector benefits from knowing the natural output distribution of the suspect model in the absence of a watermark, enabling more precise detection of infringing behavior.

---

> ### Author Response · Authors · 2024-11-24
> **Thanks for your invaluable comments and advice. (2/2)**
>
> #### **Weakness 3: Impact of Same/Difference Source/Suspect Base Model.**
>
> In our experiments, we use llama2-7b and llama3-7b as source models and bloom-7b and mistral-7b as suspect base models. Notably, the structure of llama2-7b and llama3-7b are different (e.g. the tokenizer of the two models are different, and the vocabulary size of llama3 is increased from 32k to 128k). We conduct experiments comparing identical source and suspect models (e.g., llama2 to llama2) with different source and suspect models (e.g., mistral to llama2) under Unigram watermark and Code training data settings. The results are summarized in the table below.
>
> | Method | Source LLM | Suspect LLM | TPR | TNR | ACC | DSR |
> | --- | --- | --- | --- | --- | --- | --- |
> | LIDet | Llama2 | Llama2 | 1.0 | 1.0 | 1.0 | 1.0 |
> |  | Mistral | Llama2 | 1.0 | 0.9 | 0.95 | 0.9 |
> | Baseline | Llama2 | Llama2 | 0.7 | 0.7 | 0.7 | 0.4 |
> |  | Mistral | Llama2 | 0.9 | 0.6 | 0.75 | 0.5 |
>
>
> The experimental results show that using the same base model for the suspect model as the source model increases watermark detection rates. This is particularly evident when the suspect model also has its base model used as an anchor model. Additionally, when employing mistral as the source model and llama2 as the suspect base model, the proposed method outperforms the baseline, achieving a notably high detection accuracy even with differing source and suspect base models.
>
>
> #### **Question 1: Comparison between LIDet and [1].**
>
> Compared to the method proposed in [1], our approach adopts a more realistic threat model, where both the suspect base model and its training data are unknown. This introduces greater challenges for watermark detection, rendering the method in [1] less applicable in such scenarios. Additionally, in i.i.d. detection, our method demonstrates higher stability. Specifically, in cases involving the Unigram watermark, the green ratio mismatch significantly affects the detection accuracy of [1], whereas our approach addresses this issue effectively, achieving superior performance.
>
> #### **Question 2: Should the set of anchor models always contain the same source model architecture?**
>
> The Anchor Model does not necessarily have to include the source model, but doing so is a straightforward choice since the detector can ensure the source model is completely unwatermarked. Due to the generalization capabilities of LLMs, various anchor models can sample text that aligns with natural language green ratios. Thus, the effectiveness of this method is not highly sensitive to the specific anchor model chosen.
>
> #### **Question 3: Impact of the suspect model's dimension.**
>
> Intuitively, the ability to detect infringement correlates with how well the suspect model learns the watermark. Compared to factors directly influencing model training, such as learning rate or the number of training steps, the size of the suspect model has a relatively minor impact on watermark detection (e.g., Table 6d in [1]). Furthermore, when owners of similarly sized models use larger datasets for instruction tuning, the watermark's detectability may actually increase due to the more extensive training.
>
> #### **Question 4: Correlation between detectability and domain of training queries**
> The domain of the data used to train the suspect model significantly impacts both the learning and detection of the watermark, thereby affecting its detectability. For example, when code-related queries are used for training, the sampled training data from the source model is limited to a code-specific distribution. Consequently, watermark tokens learned by the suspect model are often task-related (e.g., variable names, and comments). If the detector queries the suspect model using a domain with a different distribution (e.g., law or medical), the sampled data contains fewer watermark tokens, reducing detectability. Thus, our detection method aims to obtain a wider range of tokens and a more general distribution from the suspect model, enhancing watermark detection capabilities.
>
>
> [1] Tom Sander et al. Watermarking Makes Language Models Radioactive, NeurIPS 2024.

---

> ### Author Response · Authors · 2024-11-28
> **Request for Your Invaluable Feedback**
>
> Dear Reviewer RZ9s,
>
> We are deeply grateful for your valuable comments and suggestions and we would like to know whether our rebuttal has addressed your concerns properly. We sincerely request your further feedback, as your insights are invaluable in helping us improve the quality of our work.
>
> Best regards, Authors

---

> > ### Comment · Reviewer_RZ9s · 2024-12-02
> > **Ack**
> >
> > Dear authors, I appreciated your response and additional experiments. I'll thus increase my score.

---

### Meta-Review · Area_Chair_thVx · 2024-12-09

**Metareview:**

The paper presents a novel method called LIDet to detect whether an LLM has been fine-tuned using the outputs of another LLM, which is an interesting question. According to the reviews and the authors' rebuttal, I believe the issues raised by the reviewers can be effectively addressed. In light of the overall positive reception, I recommend accepting the paper.

**Additional Comments On Reviewer Discussion:**

Reviewers raised concerns about the lack of comparison with other methods, the absence of a comprehensive evaluation, and the transferability of the approach. The authors have provided additional results and clearer explanations to address these issues, which is approved by most reviewers.

---

### Decision · Program_Chairs · 2025-01-22

Accept (Poster)